# Therapy for Temporomandibular Disorders: 3D-Printed Splints from Planning to Evaluation

**DOI:** 10.3390/dj11050126

**Published:** 2023-05-08

**Authors:** Andrea Somogyi, Dániel Végh, Ivett Róth, Tamás Hegedüs, Péter Schmidt, Péter Hermann, Zoltán Géczi

**Affiliations:** Department of Prosthodontics, Faculty of Dentistry, Semmelweis University, 1085 Budapest, Hungary

**Keywords:** temporomandibular disorders, TMD, temporomandibular joint, TMJ, additive manufacturing, occlusal device, oral splint, 3D printing, occlusal device

## Abstract

Introduction: This article describes the authors’ digital workflow-based method for fabricating intraoral occlusal splints, from planning to the evaluation phase. Materials and Methods: In our protocol, first, we had a registration phase. This included taking digital impressions, determining the centric relation (CR) position with the deprogrammer Luci Jig, and using the digital facebow for measuring the individual values. The laboratory phase was next, which included planning and manufacturing with a 3D printer. The last phase was delivery, when we checked the stability of the splint and adjusted the occlusal part. Result: The average cost is lower for a fully digital splint than for conventional methods. In terms of time, there was also a significant difference between the classic and digital routes. From a dental technical point of view, the execution was much more predictable. The printed material was very rigid and, therefore, fragile. Compared to the analog method, the retention was much weaker. Conclusion: The presented method permits time-efficient laboratory production, and may also be performed chairside in a dental office. The technology is perfectly applicable to everyday life. In addition to its many beneficial properties, its negative properties must also be highlighted.

## 1. Introduction

Since the first introduction of the Cerec system in the early 1980s, computer-aided design and manufacturing technology (CAD-CAM) has spread widely, not only in the field of adhesive restoration, but also in every field of modern dentistry. Thanks to this innovative technology, it has been possible to conduct chairside restorations fully managed by the clinician, with the advantages of lower costs, more rapid production and the exclusion of the provisional phase [1].

One of the key components of digital dentistry is 3D printing, and it is expected to grow rapidly [2,3]. As a result, dentists need to learn relevant information to help them to incorporate this technology into their everyday practices. Despite the fact that these additive inventions are already widely used, 3D printing is not introduced to most dentists during their undergraduate education [4]. However, the most recent generation of dentists possesses the necessary skills and fundamental knowledge of digital dentistry and can be quickly introduced to the new workflow provided by 3D printing, demonstrating their willingness to invest in this field [5]. The updated digital workflow also improves the patient experience by allowing CAD solutions to visualize anticipated outcomes in new ways [6]. For dental students, the digital process is not a problem. They become familiar with the essential procedures of digital dental treatment during their education, such as digital impression taking, additive manufacturing and intraoral scanning. The potential for digital dentistry to develop is vast among this new generation of dentists. Because the patient could use digital dental tools to follow their entire treatment, the workflow and patient outcomes could be created together. Using social media is crucial [7]. It is currently having a major impact on the healthcare industry and is a great resource for facilitating expert knowledge and experience sharing. Additionally, it appears to be a quicker method than official support networks for obtaining assistance if needed.

### 1.1. Etiology of Temporomandibular Disorders

With further improvements in chairside technologies and materials, specifically in 3D printing, an oral splint to treat temporomandibular disorders may at present be created in only one day. Temporomandibular disorders (TMDs) are a collective term for musculoskeletal conditions that could affect the temporomandibular joints (TMJ), masticatory muscles and related structures [8]. According to some estimates, up to 40% of individuals living with TMD symptoms will experience spontaneous recovery of their complaints [9]. Sounds in the TMJ and deviation when opening the jaw are common (about 50% of the population) and considered to be normal, so they do not require any treatment. There are more concerning signs and symptoms, such as reduced mouth opening and occlusal changes (which affect approximately 5% of the population) [9]. TMDs have a considerable prevalence and substantially impact physical and psychosocial factors [10]. The major symptoms of TMDs include pain or tenderness of the jaw; pain in one or both of the temporomandibular joints; aching pain in and around the ear; difficulty or pain while chewing; and aching facial pain or locking of the joint, which makes opening and closing the mouth difficult [9,11].

### 1.2. Treatment Options of Temporomandibular Disorders

TMDs also cause considerable socioeconomic costs, which are usually caused by other health problems, such as depression and other psychological issues [12]. The suggested treatments for TMDs vary over an incredible spectrum of modalities. At present, we still do not have a perfect treatment solution for every patient. The reason for this is their multifactorial etiology. The clinician has numerous recommendations for treatment methods, from patient education to surgical interventions. There are two main types of treatment modalities: conservative (reversible) (such as biofeedback, oral splints, physiotherapy and patient education) and non-conservative (irreversible) (such as arthroscopy, orthodontic treatment, TMJ surgery and irreversible occlusal treatment). Because of the lack of evidence-based treatment modalities, it is recommended to start with conservative solutions and later to move on to non-conservative solutions. To accurately assess occlusions, the condyle must be guided into the proper position. Before evaluating the occlusion, the condylar position should be examined if there is an occlusal discrepancy between the position of the condyle in the intercuspation position (ICP) and centric relation (CR). Occlusions in ICP and in CR differ for patients with an unstable occlusion. Joint sounds or pain in the temporomandibular joint (TMJ) may be a sign and symptom of this occlusal imbalance between the ICP and CR. To manage TMJ sounds and pain, occlusal correction or orthodontic treatment may be potential options, so determining the patient’s precise occlusion is crucial for treatment planning. For this reason, a facebow transfer and an articulator should be used to ascertain the dynamic and static positions of the mandible. To assess mandibular movement, an accurate depiction of the condyle and its axis–orbital plane on the articulator is necessary. The orbital point serves as the hinge axis’ anterior reference point, and the condyle’s medial pole serves as its posterior reference point in the axis–orbital plane. For transferring this information to the articulator, a facebow transfer is crucial. Although using an articulator and a facebow transfer can be complicated, the axis and orbital point of the condylar hinge must be accurately transferred to the articulator. Furthermore, it is still not clear whether the procedure can be repeated.

At present, virtual facebow transfer and virtual articulator mounting are possible thanks to the development of digital technology. The expertise of the treating healthcare provider may largely influence the selection of a treatment modality [13]. Oral splints or oral appliances are modern methods to manage temporomandibular disorders. Most of the time, using occlusal splints is a non-invasive, reversible way to help dealing with the symptoms of TMDs [14,15].

Splint therapy is the foundation and an essential component of any TMJ treatment plan. For many TMJ conditions, it is the primary therapeutic device. Splints can be used to stabilize the bite, treat temporomandibular disorders or protect the teeth from damage and wear [16]. The protection of the TMJ discs from dysfunctional forces that can cause perforations or permanent displacements is a common goal of occlusal splint treatments. Other treatment goals include improving jaw–muscle function and relieving associated pain by establishing a stable, balanced bite [16,17,18,19].

The conventional process of preparing an occlusal splint requires several steps. In brief, we need an adjustable articulator, a facebow, equipment for registering the CR position and a dental technician to make an acrylic splint. To reach our goal of using splint therapy to reduce the pain caused by temporomandibular disorders, we need more clinical appointments [20,21].

However, evidence-based guidelines for the storage media of CAD-CAM fabricated occlusal appliances are lacking. This article presents a way to prepare an oral splint for a patient living with a TMD. We present fully digital methods to manufacture an oral splint in one day. In this process, we do not need a dental technician [22].

## 2. Materials and Methods

### 2.1. Intraoral Scanner, Digital Facebow, 3D CAD Design and 3D Printing

Equipment:Trios 3 wireless intraoral scanner (3Shape Unite, software version 21.4, Copenhagen, Denmark);Zebris for Ceramill digital facebow device (software version: 1.2.4, Amman Girrbach, Pforzheim, Germany);Self-designed anterior flat plane deprogramming appliance (Lucia Jig);Zebris Ceramill Mind computer-aided design software (version: 4.2, Amman Girrbach, Pforzheim, Germany);NextDent 5100 (software version: 2.13 NextDent, Soesterberg, Netherlands);NextDent Ortho Rigid print resin (NextDent, Soesterberg, Netherlands).

### 2.2. Step by Step Workflow of the Fully Digital Process

Our fully digital process was divided into three stages (Figure 1): a registration phase involving intraoral scans and digital facebow and centric relation (CR) registration; a laboratory phase, involving splint design and 3D printing; and the delivery phase, involving the final adjustments.

#### 2.2.1. Registration Phase

The first step in making a digitally prepared oral splint was to obtain a digital impression of both the mandibular and maxillary arches. This clinical step took only 15–20 min with our Trios 3 basic wireless intraoral scanner. Following the digital impressions, the centric relation position of the mandible was recorded. We used the self-designed deprogrammer Lucia Jig (Figure 2a). Using silicone bite registration material (Prestige Bite, Vannini, Dental Industry, Grassini, Italy), the anterior deprogrammer appliance (Lucia Jig) was placed on the upper incisors (Figure 2b).

The Lucia Jig appliance provides a platform for the patient’s incisors, on which they occlude, while the posterior teeth remain out of contact, thus relaxing the muscles [23]. The patient is instructed to gradually occlude the lower central incisors on the occlusal plate until the posterior teeth are separated by about 2–2.5 mm. The vertical dimension of the Lucia Jig was adjusted in the mouth until there was only one contact point between the palatine wedge vertex and the lower central incisors, resulting in a maximum disocclusion of 2–2.5 mm between the posterior teeth [24]. The deprogramming process took approximately 30 min. Bite registration was conducted with the intraoral scanner while the mandible was in centric relation position with the anterior deprogrammer appliance. During this process, the vestibular surface of the molar teeth on the mandible and maxilla must be scanned. After the bite scan, the 3Shape intraoral scanner software fitted the upper and lower jaws together.

Following the intraoral scan phase, we applied the Zebris to record the individual TMJ values and the position of the maxilla for the CAD. The last step of this registration was the scanning of the upper bite fork of Zebris (Figure 3). To obtain the position of the upper jaw in the lab software, we used an intraoral scanner to scan the upper bite fork. For the CAD software to be able to read the scan files, they must be exported in the STL format. Additionally, we sent the STL files to the lab via email. All the data were transferred to the design software Ceramill Mind (Amman Girrbach, Pforzheim, Germany), where they were integrated. In the software, the oral splint was designed.

#### 2.2.2. Laboratory Phase

The oral splint was designed using the computer-aided design software Ceramill Mind. After obtaining the digital impression and bite scan in the right format, we also needed to send the jaw motion file from Zebris to Ceramill, which contained all the information about the patient’s jaw values. After importing the files, we were able to start the design process of the splint. Several factors were considered during the design phase. First, we had to select the upper or lower jawbone. Therefore, first, we must choose which jawbone we want to work on. Next, we defined the margin line. On the buccal side, we defined it below the equator, while on the palatal side, a few millimeters of gingiva were covered. This margin line design can significantly improve the retention of the splint, and the program verifies the splint’s insertion direction. If there were more significant changes in the direction of an axis, the program suggested a flexible margin line design.

The software also recommended a minimum splint thickness of 0.4 mm. The distance from the splint surface was shown by the color scale in the design software (Figure 4). The smoothing command achieved a good surface quality for the three-dimensional splint model. Smoothing, which removes sharp corners and self-locking forms, was also used to remove the too-tight fitting on the teeth [25].

On the occlusal surface, slight contact was made with each tooth in the “intercuspal position”. This was particularly important because it was necessary to prevent the teeth from elongating when the splint was used for a longer period of time. We designed a splint that provides canine guidance. Depending on the types of occlusion and tooth axis deviations, the design process takes about 30–60 min. The completed oral splint design was printed using a 3D printer, in our case, a NextDent 5100 (NextDent, Soesterberg, Netherlands) with NextDent Ortho Rigid print resin. This printer used a non-contact membrane Digital Light Printing (DLP) method (Figure 5).

The support structure was the added component that carried the overhanging structure or bridge structure during splint slicing, and must be removed after printing. After the splints were removed from the 3D printer, the post-processing procedure started. The freshly printed resin splint required a two-stage soaking in isopropyl alcohol to remove the unpolymerized parts of the printed product.

The LC-3D Print Box was used to dry and polymerize the 3D-printed splints that needed an additional polymerizing process. This unit was equipped with 12 pieces of 18-watt lights. Following these steps, the item must be dried and completed with a final polish.

#### 2.2.3. Delivery Phase

After the laboratory phase, we placed the final splint in the patient’s mouth. During the evaluation, it was necessary to check the fit of the splint. This included checking its rocking, the applicability of the rail and the retention. Afterwards, it was necessary to adjust the occlusion, for which we used articulation paper, a straight piece and a dental bur.

## 3. Results

### 3.1. Time

During the same appointment, it is possible and helpful to conduct both intraoral scans and digital facebow registration. In this case, it is possible to make a one-day splint.

The making of the oral splint, which is made in the dental laboratory with conventional techniques, takes ~300 min. In the case of the digital process we described, this time was much shorter, at only 135 min: ~15 min of the preparation of the materials and process, approx. ~90 min of printing time and ~30 min of post-processing and finishing.

In our process, the treatment time with the patient to make a digital impression, bite registration and digital facebow recording took approx. 60–75 min.

The printing time in the 0° and 30° orientation took ~30 and ~60 min, respectively.

### 3.2. Measurement of CR and Individual Values 

The primary goal for the mandible is to be in the CR position when wearing the splint. Various clinical techniques are available for registering the mandible in the centric relationship. The Lucia Jig utilizes the anterior stop to help to find the centric relation position [26]. This straightforward technique helps to preserve the minimal interocclusal distance for the minimal splint thickness.

The Zebris for Ceramill has a JMA optical sensor system. With this device, we recorded the individually dynamic movements of the mandible and the position of the maxillary dental arch as a digital facebow. This information was transferred into the CAD software very easily and quickly.

### 3.3. Planning

Because Ceramill Mind does not work with the original 3Shape file, STL data transfer was the first step from the 3Shape Unite.

However, our experience showed that this minimal thickness needs to have sufficient strength and durability to withstand powerful occlusal forces. In addition, we observed more damage (cracks) after several uses. We, therefore, established a minimum thickness of 1 mm.

This software has a disadvantage, as it imitates the movement of the mandible according to the values recorded by Zebris, but it is not able to follow the constructed surface of the splint. For example, during canine guidance, when the canine “goes” into the designed virtual splint, the program indicates its extent with a color scale. With this function, we cannot perfectly evaluate the exact process by which the constructed surface guides the mandible.

The Ceramill Mind software also has a disadvantage: it imitates the movement of the mandible according to the values recorded by Zebris, but it is not able to follow the constructed surface of the splint. For example, during canine guidance, when the canine “goes” into the designed virtual splint, the program indicates its extent with a color scale. With this function, we cannot perfectly evaluate the exact process by which the constructed surface guides the mandible.

### 3.4. Three-dimensional Printing

The machine had an additional setting for printing direction ranging from 0 to 90°. It was crucial because the printing direction affects the printing time and the amount of splint material used [27]. Using the 3D printer with the least amount of material is needed for the 90° range because it requires fewer supports to hold the splint. The 0° printing setting required the least amount of printing time, which is also a good option. The literature recommends a 30° orientation for aligners and surgical splints [28].

Despite printing the same STL file, the splint did not fit in the patient’s mouth every time or was too tight. In most cases, the 0° and 30° orientations were the best.

### 3.5. Evaluation

First and foremost, we needed to double-check the fit, which the material’s transparency made simple. Because of the transparency, it was possible to determine whether it fitted perfectly in all areas. The second step was to evaluate the stability and retention of the splint. After that, the occlusion was adjusted, which was faster with the 3D-printed splints than with traditional methods in our experience.

### 3.6. Wearing Splint Experience

Much less follow-up work is required when using the digital pathway, contrary to the traditional process. The average delivery time for conventional splints is 30 min, while for 3D-printed splints is 15 min. With the same material thickness, 3D-printed splints break much more easily.

## 4. Discussion

In the field of dentistry, digital technology is developing very quickly. There is an increasing demand for the most efficient use of expensive digital equipment. More and more dental practices are using digital scanners and digital facebows. It is an excellent option to improve splint therapy in temporomandibular disorders. We anticipate that dental offices and dental laboratories will rapidly adopt 3D-printed orthodontic appliances. The benefits are its accuracy and a light workload. It is precise, long-lasting, less expensive and quicker than the conventional method [28,29].

The following presents some aspects that we would like to highlight based on our experience with fully digital splint therapy:

### 4.1. Materials

The manufacturer’s material for a 3D printer is recommended for use in printing oral splints. There is other available material for oral splints. A recent study found that the cytotoxic effects of resins for 3D-printed, milled and regular oral splints were almost the same [30]. At present, it is not always strictly necessary to print a master model at all; this means less use of dental materials [4,31].

### 4.2. Cost

A fully digital process necessitates the use of costly software and hardware components. However, if this investment has already been made, the cost of printing splints is less than that of using the conventional method.

It is possible to rescan a splint that has been individually adjusted in the patient’s mouth. If it is necessary to reprint the splint, we could print the finished (mouth-adjusted) version. We see a significant advantage when we have all the equipment (including the 3D printer); we can plan and prepare the oral splint in our dental office because the software is very user-friendly, so we can also train our practice staff in the process. Therefore, it can decrease our costs. Shortening the workflows performed by dental technicians can help them to spend more time on other dental processes, thus increasing their performance and cost efficiency.

### 4.3. Planning

In the literature, we did not find any clear indication of the orientation in which the splint should be placed in the 3D printer in order to achieve the most accurate results. In some cases, the 30° position was mentioned [27,32]. For this reason, in several cases, our laboratory produced the designed splint in the 0°, 30° and 90° orientations. During the placement in the mouth, we experienced large differences based on the patients’ feedback and our own observations. Despite printing the same STL file, the splint did not fit the patient’s mouth or was too tight. In most cases, the 0° and 30° orientations were the best. This discovery is significant, as the 90° orientation is not recommended for dental laboratories. The best performance of the printer, however, would necessitate the 90° position.

### 4.4. Time

Two main work phases can be identified: 1. The dentist or dental technician takes the necessary steps. These phases depend on the experience of the dentist and dental technician, the type of scanner, the digital facebow and the software. There have been several studies on the scanning time of dentists in which their learning curve was recorded. The results showed that the plateau phase was reached at about 15 digital impressions [33]. This result agrees with the learning process’s general results. Accordingly, this is the approximate number of measurements that should be expected when using a digital facebow. In addition to the learning curve of the dental workflow, the active feedback of the practicing dentist plays an important role. In our process, the treatment time with the patient to make a digital impression, bite registration and digital facebow recording took approx. 60–75 min.

The printing time depends on the type of 3D printer and the software settings. The location of the splint on the printer tray also has a vast effect on how long it takes to print. In our case, if the splint was placed on the tray at 0 degrees, the printing time was ~30 min; at 30°, the printing time was ~60 min; and at 90°, the printing time was ~90 min (for an average arch size).

### 4.5. Sustainability

Except for the supports, there is no material loss in additive technology. This contrasts with subtractive technology, where the product is milled from an existing block, which creates a significant environmental burden and a waste of material that cannot be recycled. Digital technology could have the advantage of transforming workflows. Certain steps can be omitted, such as the production of dental models [34].

### 4.6. Wearing Splint Experience

In this field, we could see some deficiencies. In our experience, the resin materials are strong and hard, but a little bit too hard, which cause some discomfort when worn. Unfortunately, the splint material is incapable of elastic deformation, so it can be easily broken under higher loads, such as in patients with bruxing habits. This negative effect was most noticeable when correction was required in the molar region after 3D printing because the planned optimal thickness of the splint had to be thinned out. This could be the reason why we experienced breakage in several cases.

### 4.7. Future Perspective

This digital splint-making process requires the procedure to be dynamic, which means that it is overall faster and cheaper than the conventional splint-making process that is currently being used. At the moment, this is the case if the dentist is digitally qualified. This makes it possible for an average dentist to use this digital method on a daily basis, which requires a significant financial investment but shortens the amount of time needed to treat patients.

We hope that this method will be a key component of a future clinical study that we plan to conduct with a larger patient population.

## 5. Conclusions

In summary, we prepared oral splints with a fully digital method that has many advantages. First of all, it has a short working time and an easy standardization of the work process; it is also a much cleaner and faster process than conventional splint methods. All work processes can be conducted in a properly equipped clinic, but this requires a large financial outlay. Unfortunately, there are also some problems. For example, the accuracy of the printing depends on how it is oriented (0, 30 or 90 degrees), the splints can be hard to keep in place or the printed material is too rigid. We are already utilizing this technology, so we would be delighted to assist our students in developing their digital manufacturing projects for splints and other medical and dental items.

## Figures and Tables

**Figure 1 dentistry-11-00126-f001:**
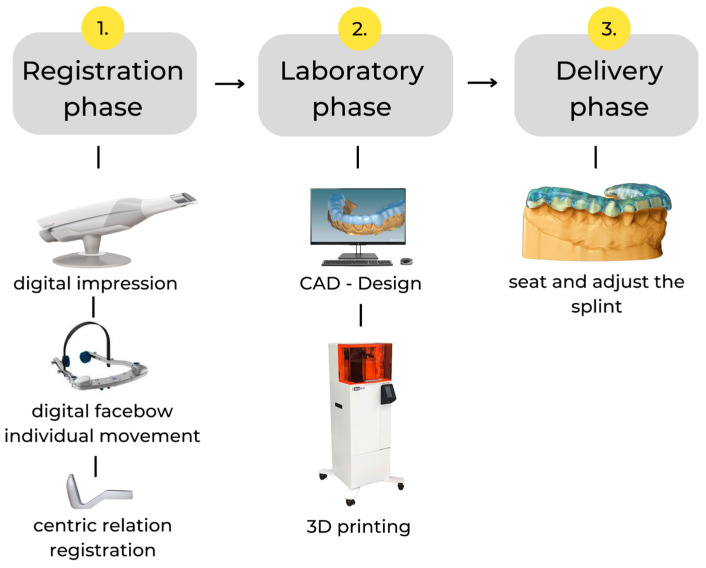
The fully digital process of manufacturing a 3D-printed splint.

**Figure 2 dentistry-11-00126-f002:**
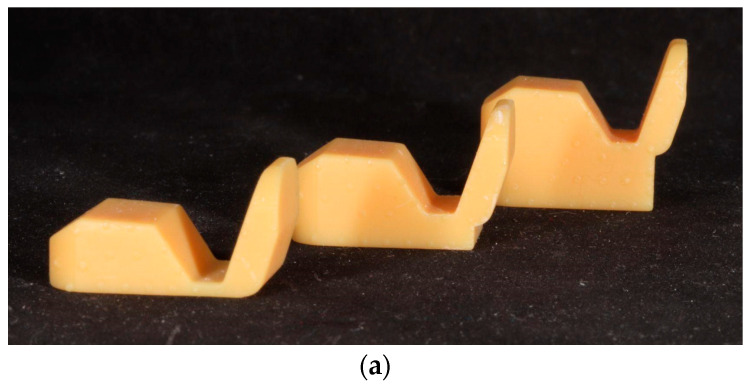
(**a**) Self-designed Lucia Jigs with different vertical height; (**b**) centric relation registration with Lucia Jig.

**Figure 3 dentistry-11-00126-f003:**
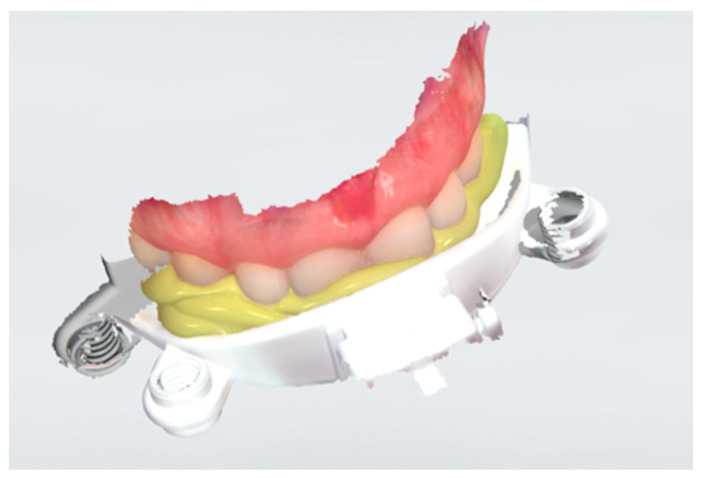
Scanned upper bite fork of the digital facebow.

**Figure 4 dentistry-11-00126-f004:**
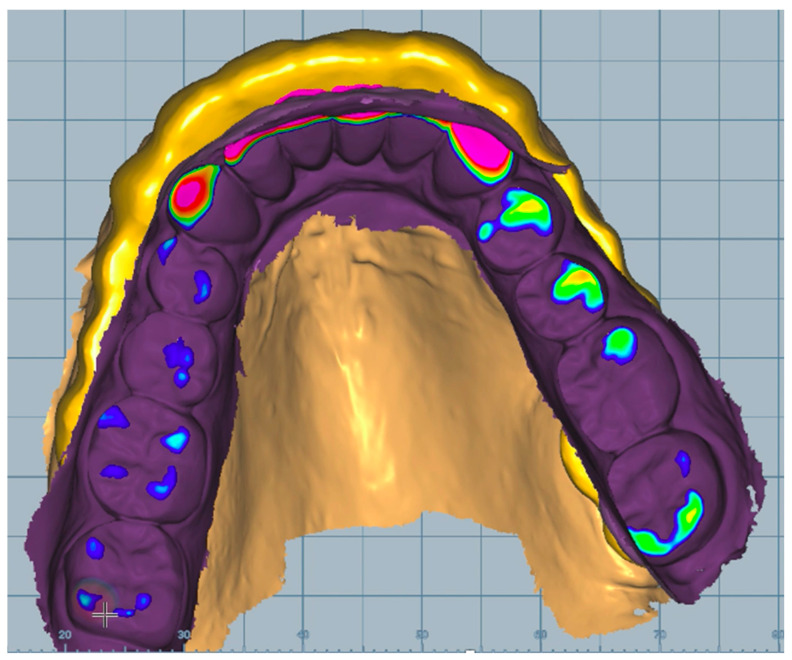
Design of the splint in the CAD software.

**Figure 5 dentistry-11-00126-f005:**
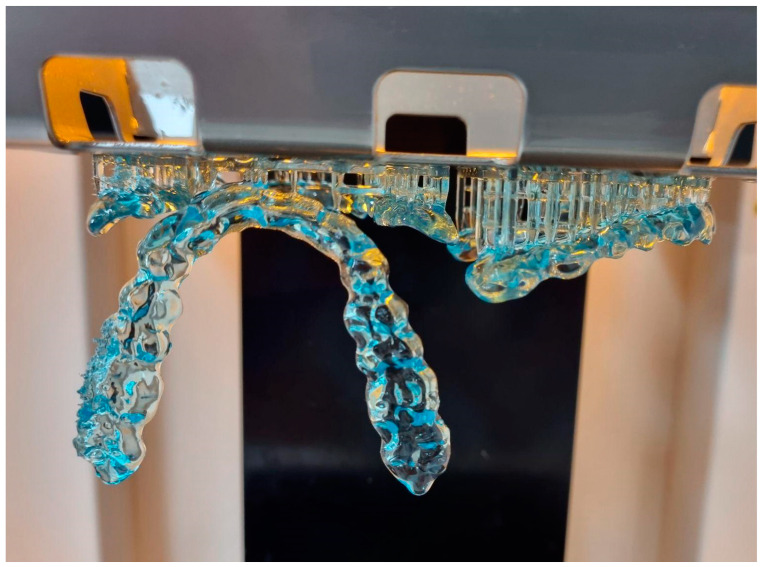
Printed splints in 0°, 30° and 90° orientations (Nextdent 5100).

## Data Availability

The data presented in this study are available upon request from the corresponding author. The data are not publicly available due to privacy matters.

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
