# Peer review of "Therapy for Temporomandibular Disorders: 3D-Printed Splints from Planning to Evaluation"

_dentistry, 2023, doi:10.3390/dj11050126_

Round 1

Reviewer 1 Report

This technical report is interesting and well presented. Some technical details/images could be added to improve the manuscript. However, a clinical trial is needed to compare this method with the conventional method in terms of retention, efficacy, timing, accuracy etc. Therefore, we can't conclude that it is better than the conventional method, instead we can conclude that it might be better in some aspects (this point has to be considered in the manuscript writing).

Author Response

Thank you very much for your work; we have corrected the grammatical errors. In terms of both the passive structure and the tense.

Regarding the length of the paragraphs, the newspaper has rules regarding the minimum length. The text was formulated accordingly.

Looking at the paragraphs, we tried to describe the topic as clearly as possible. In addition, we tried to maintain the proportions between the parts of the article.

We did extensive language editing of English language and style, and we transferred the subjective findings from the methods section to the result section, so the proportions between the introduction and the result have already been restored.

Reviewer 2 Report

The article entitled: “Therapy of temporomandibular disorder: 3D printed splints 2 from planning to evaluation” may be a relevant contribution to the literature, however many critical issues must be addressed first.

1.      All along the manuscript, some recurrent mistakes can be found:

a)      There is a confusion about verb tense: sometimes in present tense, sometimes in past tense. It should be in past tense.

b)     Sentences should be on passive voice (impersonal)

c)      Too long paragraphs.

2.      Abstract is not the right place to include brands/models. More precise terms should be used instead of: “a few times” (L-21)

3.      In the Introduction section a very long paragraph (L-36 to L-53) make statements about: “the new generation”, “social media”, “facilitating expert knowledge”. No reference to support these statements is provided by the authors. It must be provided.

4.      Item 2.1 in Materials and Methods presents a list of equipments, makes and models. I suggest including these equipment details along with the description of the procedure.

5.      The limitations (advantage/disadvantage) of the methods/equipments should be done in Discussion section if so, but not in Materials and Methods section.

6.      As far as a patient is involved, an Ethical Committee approve protocol must be provided.

Author Response

1.Thank you very much for your work, we have corrected the grammatical errors. In terms of both the passive structure and the tense.

Regarding the length of the paragraphs, the newspaper has rules regarding the minimum length. The text was formulated accordingly.

Looking at the paragraphs, we tried to describe the topic as clearly as possible. In addition, we tried to maintain the proportions between the parts of the article.

We made an extensive language editing of English language and style and we transferred the subjective findings from the methods section to the result section, so the proportions between the introduction and the result have already been restored.

2.We removed the brand names from the abstract and corrected the mentioned parts.

Patient Consent Form for Articles Containing and ethical permission were subsequently sent to the editorial office. We added this to the text of the article.

We have corrected the mentioned inaccuracy.

3. The introduction was really long compared to the result part. In accordance with the recommendation of another reviewer, we transferred the subjective findings from the methods section to the result section, so the proportions between the introduction and the result have already been restored. Some references to the statements have also been added to the mentioned paragraph.

4. Thank you for your comment, the tools are also marked when describing the workflow. Basically, we chose them separately at the beginning of the chapter so that the reader can see in one place what tools we used, and not have to pick them out one by one from the text. But we have changed the title of chapter 2.1 to make it clearer.

5. This is a very good insight, thank you very much! Accordingly, the mentioned text was transferred from the methodology section. As a result, the article became much more understandable and transparent!

6. After the article was uploaded, the patient's consent statement and ethical permission were also uploaded. It was also written into the text of the article.

Reviewer 3 Report

This article describes the authors' digital workflow-based method for fabricating intraoral occlusal splints, until the evaluation phase. Unfortunately, the described workflow does not introduce an original method, the scientific documentation is poor and the authors cite most of the time their opinion, the results are not sustained by tests and validation.

I recommend the authors:

1.      to restructure the material as a technical report (where a step-by-step presentation of the method is recommended) or as a clinical study ( where the study has to answer a central question, results must present a standardized method to prove the validity of the technique and discussions where  a comparison with other methods must be approached)

2.      to clarify how they have checked the proper transfer of the mandible-maxillary relation in the design soft- the automatic procedure has many errors.

3.      to have a control to report at - an analog splint method

4.     to include relevant tables and figures.

5.      to sustain their opinion with relevant references

Author Response

Thank you for your work and review, we have improved the article based on what you wrote.

  1. Since this is not about the presentation of a case (despite the fact that the methodology was part of the case of a patient presented with pictures), as a case report it does not correspond to what is described by the journal. Not even a review. Since we are describing experiences from several cases, we decided on the original article for this reason.

In the article, we describe the methodology of making a splint, with which we standardize our subsequent scientific investigations.

At the same time, we want to provide our readers with clinically useful experiences. The authors of the article miss these in the articles published on the subject.

This is a pilot study on our part, in which we want to create a solid foundation for our own research colleagues and colleagues at other universities. To describe our practical experiences with which we can help their work. Also, we can help them avoid making the mistakes that we might have made.

  1. Basically, based on our practical experience, there were no problems in determining the position of the mandible and maxilla in the Trios system. The development of digital technology is the result of software updates and a lot of experience, so that now the bite scanning is no longer behind in accuracy compared to the analog method. (https://www.ijoms.com/article/S0901-5027(23)00048-6/fulltext#secsect0065)

Moreover, when there is an interocclusal space, the software assembles the digital samples more accurately than when there is no space. (https://www.thejpd.org/article/S0022-3913(22)00564-9/pdf)

In the case of inexperience, there is a high chance of inaccuracy with both the analog and digital methods.

Articles have also been published regarding the accuracy of the digital facebow, which state that there is no statistically detectable difference between the analog and digital methods. https://pubmed.ncbi.nlm.nih.gov/35382941/

  1. The analog and digital routes were compared at several points. But introducing the analog route would have involved a lot of extra text. Many types can be distinguished among the traditional routes. If we had explained all this, the article would have been unrealistically long and attention would have been diverted from the original message. In these cases, we marked the difference in each case.

  1. We included the amount of images required by the magazine in the article. Tables cannot be prepared in this study, as the results were not of the type that can be presented in a table.

  1. New references have been added, thanks for the comment. The conclusions we have drawn are new results, for which we cannot attach a reference, since they have not yet been described in this form in the literature.